# Boron Nutrition in Coffee Improves Drought Stress Resistance and, Together with Calcium, Improves Long-Term Productivity and Seed Composition

**Victor Hugo Ramirez-Builes** [1,*], **Jürgen Küsters** [1], **Ellen Thiele** [1] **and Luis Alfredo Leal-Varon** [2]

1   Center for Plant Nutrition and Environmental Research, Yara International, 48249 Dülmen, Germany
2   Yara Colombia, Bogotá 130009, Colombia
*   Correspondence: victor.ramirez@yara.com; Tel.: +49-025947980 (ext. 112)

**Abstract:** Coffee production around the world is under threat due to climate variability and change, and drought stress will increase in many coffee regions; nutrient management practices can enhance the adaptation capacity of coffee production. Considering that coffee is a crop sensitive to boron (B) deficiency, this research investigated how B nutrition improves resistance to drought stress in coffee under controlled conditions as well as how the interaction with calcium ($Ca^{2+}$) influences productivity, seed composition, and soil fertility during a production cycle of 5 years. Under controlled conditions for seven months, coffee plants were submitted to water stress with and without B nutrition. In the field, the coffee plantation was exposed to two B rates (0.6 and 1.1 kg.ha$^{-1}$-year$^{-1}$) with a fixed calcium ($Ca^{2+}$) rate and a control without $Ca^{2+}$ and B. After 9 months of growth and seven months of water deficit, under controlled conditions, the application of B significantly ($p < 0.05$) reduced the negative effect of water stress on coffee growth. Under field conditions, after a production cycle of 5 years, the application of 0.6 kg B and 77 kg CaO. ha$^{-1}$.year$^{-1}$ yielded 14% more than the control without B and $Ca^{2+}$. An increase in the B rate to 1.1 kg.ha$^{-1}$.year$^{-1}$ with the same $Ca^{2+}$ rate increased the productivity significantly ($p < 0.05$) by 37% compared to the control. The B application also improved significantly ($p < 0.05$) the B content in the soil at 0–30 cm. The B treatments influenced the biochemical composition of the green coffee beans, with a significant ($p < 0.05$) inverse correlation between the B content of the green coffee beans and caffeine and unsaturated fatty acids (UFAs), mainly, oleic, linoleic, and linolenic acids, which are considered negative descriptors of the coffee quality.

**Keywords:** coffee; water stress; caffeine; fatty acids; productivity

## 1. Introduction

Coffee is one of the most popular beverages worldwide and one of the most important traded agricultural commodities. Coffee is produced in more than 30 countries, located in tropical and subtropical areas, providing support to nearly 25 million coffee producers. Coffee farming and processing are labor-intensive and tend to be the primary source of livelihood in producer countries [1,2]. Coffee provides economic benefits at each step of the global value chain, linking growers to consumers, and contributes to the economies of both exporting and importing countries; as a beverage, it causes joy to a growing number of consumers around the world [3]

Coffee is a drought-sensitive plant with significant changes in water leaf status, photosynthesis activity, and antioxidant enzyme activity depending on the stress duration and magnitude, resulting in significant changes in the plant morphology, growth, and productivity [4–8]. Global projections show that the area suitable to grow coffee could be reduced by more than 50–70% in 2050 due to climate change, mainly by changes in the air temperature and prolonged drought periods [9,10].

Boron (B) deficiencies in crops are rather frequent and widespread [11], and an important deficiency in both acidic as well as calcareous soils [12–14]. Coffee is considered a B-sensitive and -responsive species [15]. In coffee, B is the third most absorbed micronutrient by the coffee beans after iron (Fe) and zinc (Zn) or manganese (Mn). One ton of green coffee beans absorbs between 40 and 55 g of B [16–19]. Boron fertilizer is recommended when the B content in the leaves is below 60 mg.kg$^{-1}$ and B content is considered to be very low in leaves when its content is below 45 mg.kg$^{-1}$ [20–22]. Our analysis of 16,000 leaf samples of coffee plants from Brazil, Colombia, Salvador, Guatemala, Kenya, Tanzania, Uganda, and Vietnam revealed that 29% of the samples had a B concentration lower than 45 mg.kg$^{-1}$ (Yara International, Pocklington, UK, Megalab database, data not published), clearly indicating that B deficiency should be considered as a latent threat to coffee growth, productivity, and quality. Boron deficiency may also reduce the potential of the coffee plantation to adapt to increasingly adverse weather conditions because of climate variability and change.

Boron is important in the structure of the cell wall [23], where it forms a diester bond between *apiosyl* residues and two monomers of pectic polysaccharide rhamnogalacturonan II (RG-II), forming the B-RG-II complex [23–25]. Boron deficiency alters the membrane potential and reduces the activity of proton-pumping ATPase and thus the proton gradient across the plasma membrane; B maintains the membrane structure through *cis diol* complexation with glyco-proteins, which are structural constituents of the plasma membrane. Boron influences nitrogen fixation and metabolism and controls the cell wall pores size; B plays an essential role in the reduction in the oxidative free radicals' accumulation in root and leaf cells, as well as in the reduction in reactive oxygen species (ROS) under stress conditions. Boron deficiency-induced reduced water flux via aquaporins may pose an additional threat to crops under cold or drought stress conditions, reducing the reproductive capacity and fruiting capacity through the poor formation of pollen cell walls and reducing cell expansion [24–26].

Severe B deficiency causes a rapid decrease in the amount of plasma membrane water channel proteins (ZmPIP1 aquaporins) in the root apex, lowering the ability of roots to maintain hydraulic conductivity in response to abiotic stress [26,27]. In B-deficient plants, a reduction in root hydraulic conductance may be caused by the perturbation and inhibition of growth of newly formed roots. Boron deficiency has a direct influence on the uptake of other important nutrients, like K$^+$ and solutes, into cells required for the maintenance of water uptake, cell turgidity, and expansion [28]. The knowledge on the signaling pathways for the response of plants to B deficiency has advanced greatly in recent years [29]. Chen et al. [30] made a recent review of the main process and phytohormones, like ethylene, auxin, cytokinin, brassinosteroids, and ABA, showing that Ca$^{2+}$ and ROS are involved in the orchestrated signaling pathways of B stress response.

According to Brown and Shelp [15], the pattern of occurrence of B deficiencies suggests that B deficiency in plants is physiological in nature, e.g., induced by rapid growth resulting from favorable environmental conditions and high nitrogen fertilizer levels. Coffee is a highly responsive crop to increasing nitrogen fertilization rates [31]. The response of the coffee trees to B fertilization is erratic, depending on the year, time, and application forms, as well as the B fertilizers' sources [32]. A reason for such an erratic response could be linked to the fact that trials with B in coffee are often short-term trials initiated in well-established plantations or only in juvenile plantations. For perennial crops, such as coffee, it is recommended that nutrition research be carried out through long-term experiments, encompassing a production cycle from planting or rejuvenation over 4–5 years and four harvest cycles. Visual B deficiency in coffee is commonly associated with a deformation of the youngest leaves, brown spots on the leaves, and leaf border discoloration [33]. In many crops, however, the only symptom of B deficiency may be a poor fruit set [34].

Calcium (Ca$^{2+}$) is the third most demanded nutrient by coffee after nitrogen and potassium, and in coffee, it has a demonstrated beneficial influence on productivity and stress resistance [35]. The impact of B on coffee growth, long-term productivity, abiotic

stress resistance interaction with $Ca^{2+}$, and influence on the biochemical composition of green coffee beans is not well documented. The objectives of this research were to evaluate the influence of the B nutrition on drought stress tolerance in a young coffee plantation as well as to test, in a long-term trial at the field level, its impact on coffee productivity and seed composition.

## 2. Materials and Methods

### 2.1. Greenhouse Trial

With the aim to evaluate the influence of B on growth and nutrient uptake with and without water stress, a trial was established from 2019 to 2021 under greenhouse conditions. Coffee seeds from the *Coffea arabica* var. Cenicafé 1 were pre-germinated in dark conditions with a mean temperature of 28 °C for 6 weeks using disinfected sphagnum moss as the germination medium. Before the radicle emerged (BBCH scale 03-Arcila et al. [36]), the pre-germinated seeds were moved to small containers with perlite as the growing medium. The seeds were allowed to germinate for 6 months. During this germination process, the plants received a nutrient solution once per week containing: N (7.6 mM), P (0.3 mM), K (1.7 mM), Mg (0.2 mM), Ca (0.9 mM), Fe (5.0 μM), Mn (2.9 μM), Zn (1.5 μM), Cu (0.6 μM), B (9.2 μM), and Mo (0.2 μM).

The plants were transplanted to pots of 4.5 L once they reached three leaves in completely open pairs (BBCH scale 13). After transplanting, a trial was set up to compare two treatments, with and without B application, and two water levels, with and without water stress, using a complete randomized factorial experimental design with 12 replications. For the coffee plants without water stress, the soil moisture was set to 55 to 60% of the water-holding capacity (WHC). Two months after transplanting, the plants under water stress grew at a soil moisture between 35 and 40% WHC for a period of seven months until harvest. The WHC of the coarse sand was estimated by saturating the pots with water and covering them with black plastic to avoid evaporation. Once the free drainage stopped, the pots were weighed and the moisture content at that point in time was considered as the moisture at WHC. After transplanting, the gravimetric soil moisture was measured daily using a precision scale (Sartorius Combic 1, Sartorius AG, Göttingen, Germany).

The B application was made on top of the soil using a calcium nitrate (CN + B)-based product with B (15.3% N; 18.5% Ca; and 0.32% B) in the treatment with B and the same CN product without B. The total B applied was 12.2 mg.plant$^{-1}$ split into 4 applications, with 1 application per month after transplanting. All the other nutrients were applied as a nutrient solution to the soil surface, without any foliar application. The pots were watered with a complete nutrient solution containing N (21.4 mM), P (1.3 mM), Mg (1.6 mM), Ca (0.8 mM), Fe (14.7 μM), Mn (8.4 μM), Zn (5.3 μM), Cu (14.2 μM), and Mo (1.5 μM). The nutrient solution was applied once per week with application volumes in the range of 60–120 mL, according to the water demand of the plants and the set soil moisture level of the treatment.

At the end of the trial (9 months after transplanting), the dry biomass accumulation was measured separately for roots, stems, mature leaves, and youngest leaves (the first two pairs of leaves from each branch and principal stem). All samples were dried in an oven at 65 °C until a constant weight was attained. The dried plant material was then finely ground for nutrient analysis in the lab.

The microclimate conditions of the greenhouse chamber during the trial were: Mean air temperature of 23.1 °C (±2.2 °C) with a maximal air temperature of 31.4 °C and minimal air temperature of 15.3 °C, mean relative humidity of 64% (±10%), and mean light intensity during summertime of 20.0 Klux. A supplemental light (300 μmol m$^{-2}$ s$^{-1}$ photosynthetic photon flux density) with a 12–14 h light period was provided when natural light became insufficient.

The soil and tissue samples from the field and greenhouse were analyzed in Yara's laboratory located in Dülmen, Germany. The B analysis in tissues was conducted by the microwave digestion procedure by inductively coupled plasma atomic emission spectrometry

(Perkin-Elmer 400; Perking-Elmer Corp., Norwalk, CT, USA), and N was analyzed by the Micro-Kjeldahl method. The B content in the soil was determined using a weakly buffered solution, hereinafter referred to as the CAT solution, made from 0.01 M calcium chloride solution and 0.002 M diethylenetriaminepentaacetic acid (DTPA) [37].

*2.2. Field Trial*

During five years, from July 2014 to June 2019, a coffee trial was carried out under field conditions in the southeast region of Colombia, in El Pital-Huila, at a farm located at 02°20.1′62″ N–75°50.1′41″ W and 1700 m elevation. The meteorological data during the experiments were recorded (Table 1). The soil was biotite–granite classified as a Typic Tropothents and Typic Dystrudepts [38], containing 70% sand, 24% silt, and 6% clay, with a volumetric soil moisture at saturation level ($\theta_s$) of 0.69 cm$^3$.cm$^{-3}$; volumetric soil moisture at field capacity ($\theta_{FC}$) of 0.476 cm$^3$.cm$^{-3}$, and soil humidity at wilting point of ($\theta_{wp}$) = 0.294 cm$^3$.cm$^{-3}$ at a 40 cm depth.

**Table 1.** Climatic conditions were obtained from the Simon Campos weather station (02°21′ N–75°53′ W), provided by the National Coffee Research Centre-Meteorological Network.

| Year | T. min (°C) | T. max (°C) | T. med (°C) | R.H (%) | Rainfall (mm) | Sunshine (h) |
|------|-------------|-------------|-------------|---------|---------------|--------------|
| 2014 | 15.7 | 23.6 | 19.1 | 74.6 | 1741.3 | 1233.1 |
| 2015 | 15.8 | 24.2 | 19.5 | 72.3 | 1319.6 | 1243.1 |
| 2016 | 16.1 | 24.1 | 19.6 | 73.4 | 1625.3 | 1241.4 |
| 2017 | 15.7 | 23.6 | 19.1 | 70.5 | 1976.3 | 1211.2 |
| 2018 | 15.7 | 23.5 | 19.0 | 75.3 | 1761.9 | |
| 2019 | | | | | 1482.1 | |
| Mean | 15.8 | 23.8 | 19.3 | 73.2 | 1651.1 | 1231.2 |

The field trial was established with the *Coffea arabica* L. variety Caturra growing without shade. The plantation was established in 2009 with a plant density of 6600 plants ha$^{-1}$ at a 1.5 m distance between the plants and a 1.0 m distance between rows. The plantation was stem-trimmed at a 30 cm height in August of 2014 before the treatment applications, with the aim to initiate a new production cycle. Soil samples were collected in July 2014 at a 0–30 cm depth. The analysis showed a pH of 4.4; organic carbon of 2.24%; P of 5.0 mg.kg$^{-1}$; exchangeable bases Al, K, Ca, and Mg of 244, 254, 274, and 98 mg.kg$^{-1}$, respectively; and B of 0.38 mg.kg$^{-1}$. The pH was determined in CaCl$_2$; organic matter by the Walkley–Black method; P by the Bray-II method; the exchangeable fractions of K, Mg, and Ca with 1 N ammonium acetate extraction (1 N NH$_4$C$_2$H$_3$O$_2$, pH 7.0); and B by CAT. The cations in the extracts were detected using an ICP (PerkinElmer, Optima 8300, Waltham, MA, USA), and the soil texture analyses using the Bouyoucos hydrometer method.

Previously, it was demonstrated that the optimal Ca$^{2+}$ rates for coffee in that region were between 70 and 120 kg of CaO. ha$^{-1}$.year$^{-1}$, with a mean B rate of 1.1 kg.ha$^{-1}$.year$^{-1}$ [35]. The aim of this trial was evaluate the 50% reduction in the B rate under the optimum Ca$^{2+}$ rates with respect to a control without both nutrients. With this aim, the trial was set up with three treatments: treatment 1, without Ca$^{2+}$ and B; treatment 2, with Ca$^{2+}$ and B, with mean rates of 77 kg of CaO.ha$^{-1}$.year$^{-1}$ and 1.1 kg of B.ha$^{-1}$.year$^{-1}$, respectively; and finally treatment 3, with Ca$^{2+}$ at the same rate of treatment 2, but with a lower B mean rate of 0.6 kg of B.ha$^{-1}$.year$^{-1}$ (Tables 2 and 3). Ca$^{2+}$ and B were supplied using the same calcium nitrate-based products with and without B, as in the greenhouse trail. The other nutrients (N, P, K, Mg, and S) were applied using ammonium nitrate-based NPK fertilizers.

**Table 2.** Mean nutrient application for the 5-year field trial.

| Season | N | P$_2$O$_5$ | K$_2$O | MgO | CaO | S |
|---|---|---|---|---|---|---|
| | kg.ha$^{-1}$ | | | | | |
| 2014–2016 [¥] | 251 | 84 | 186 | 38 | 109 | 35 |
| 2016–2017 | 150 | 77 | 119 | 26 | 59 | 23 |
| 2017–2018 | 250 | 90 | 240 | 49 | 52 | 56 |
| 2018–2019 | 250 | 118 | 290 | 37 | 86 | 48 |
| Average | 225 | 92 | 209 | 38 | 77 | 41 |

[¥] From stem pruning in August 2014 to the first harvest in May 2016.

**Table 3.** Mean B application for the 5-year field trial.

| B Treatments/Season | 2014–2016 [¥] | 2016–2017 | 2017–2018 | 2018–2019 | Average |
|---|---|---|---|---|---|
| | kg. B ha$^{-1}$ | | | | |
| High B rate | 0.8 | 1.1 | 1.05 | 1.39 | 1.1 |
| Reduced B rate (50% less) | 0.6 | 0.76 | 0.76 | 0.40 | 0.6 |

[¥] From stem pruning in August 2014 to the first harvest in May 2016.

The experiment was set up as a randomized completed block design with four replications. Each plot was 42.0 m$^2$ with 28 plants and 10 effective plants for yield evaluation. The harvest data were collected from January 2016 to June 2019. In this region, the main flowering period is from August to October covering 80% of the total harvest, and another short period of flowering takes place from December to March covering 20% of the harvest [39]. Due to this phenological distribution of the flowering and harvest, the fertilizer application in the experiment was split into three applications per year: the first application in August, a second application in January during the early fruit development, and a third application in March before the main harvest. This application strategy ensured proper nutrient availability for the coffee trees during the whole year.

After harvest, the pulp of the fresh cherries was removed, and the wet coffee with parchment was fermented for 24 h with a mean air temperature ranging from 18 to 26 °C, in an open non-submerged fermentation system to facilitate the degradation of the thin layer of sugars or mucilage [40]. After fermentation, the coffee was washed with fresh water and sun-dried under a plastic cover. When the coffee beans reached a moisture of 10%, the parchment was removed, and the green coffee samples were sent to the laboratory for biochemical and nutrient analyses.

In the laboratory, the green coffee bean samples were dried in an oven at 105 °C overnight and ground to obtain a fine powder. The caffeine content was determined by high-performance liquid chromatography [41], and the total lipids were extracted from 2 g samples of dried powder using the modified Folch method [42,43]. The biochemical analysis was made in the food science department of the National University of Colombia-Medellin Campus.

As the main flowering in the study region occurs between August and September and the main harvest between April and May, the effective rainfall (Pe) and the crop evapotranspiration (ETc) were estimated daily from May 2015 to May 2016 using the water balance methodology described by Ramirez and Küsters [44].

All the data were submitted to their respective analysis of variance (ANOVA) test, according to the experimental design. A statistical analysis was conducted using the Statgraphis Centurion software package Version XV (Statgraphics Technologies, Inc., The Plains, VA, USA). The normality test was conducted using a Shapiro–Wilks modified test and the heterogeneity of the variances using the residuals vs. prediction test for each of the variables. The differences among the variables were determined using Fisher´s test with an alpha value of 5%.

## 3. Results

### 3.1. Effect of Water Stress and Boron on Biomass Accumulation and Boron and Nitrogen Uptake under Controlled Conditions

In the greenhouse trial after nine months of growth and seven months of water stress, coffee growth was significantly affected by the lack of B and by water stress, resulting in a significant reduction in the shoot, root, and total biomass as well as in the root–shoot ratio (Figures 1 and 2). Under water stress conditions, B deficiency significantly reduced the total dry biomass accumulation of the coffee plants by 29%, with a reduction from 46.53 g.plant$^{-1}$ with B to 35.9 g.plant$^{-1}$ without B (Figure 1A). The differences in the total biomass between the treatments with water stress were mainly caused by the differences in shoot biomass (Figure 1B).

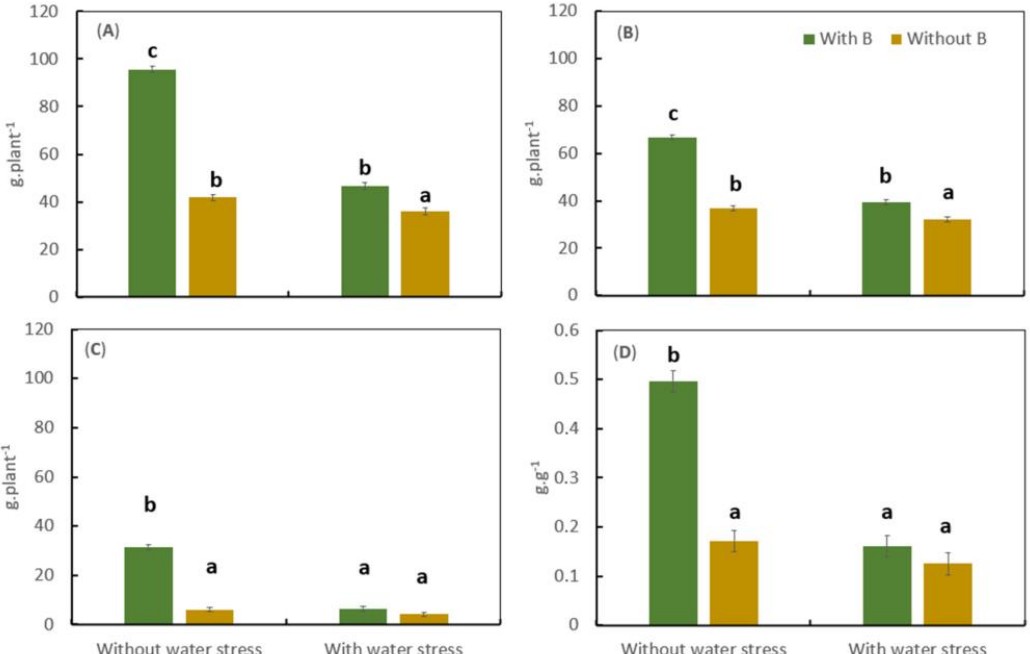

**Figure 1.** Influence of boron and water stress on coffee growth. Total dry biomass (**A**), shoot biomass (**B**), root biomass (**C**), and root–shoot ratio (**D**). Vertical bars indicate the standard error and different letters have significant differences according to Tukey's test with an alpha value of 0.05. LSD for total dry biomass = 5.5169; shoot biomass = 3.9303; root biomass = 3.991; and root–shoot ratio = 0.04528.

Without water stress, B deficiency significantly reduced the total plant shoot and root biomass (Figure 1A–C). The reduction in root biomass was more pronounced than the reduction in shoot biomass (Figure 1B), resulting in a decline in the root–shoot ratio under B deficiency (Figure 1D).

Water and B deficits significantly affected the N and B concentration levels in the leaves and N and B uptake by the coffee plants. Under water deficit, the treatments with B had 3.9 times more B in the young leaves and 3.2 times less B in the mature leaves compared to plants without water stress (Table 3). In the treatments without B, the concentration of this nutrient in the young and mature leaves did not change with the water supply level. The nitrogen concentration in the leaves was also influenced by the water level and B treatments. The plants grown under water and B deficit had significantly higher N concentrations in the young and matures leaves compared to all other treatments. The application of B significantly increased the N and B uptake by the whole coffee plants in the treatments without water stress, while under water stress conditions, the N and B uptake was not significantly different (Table 4).

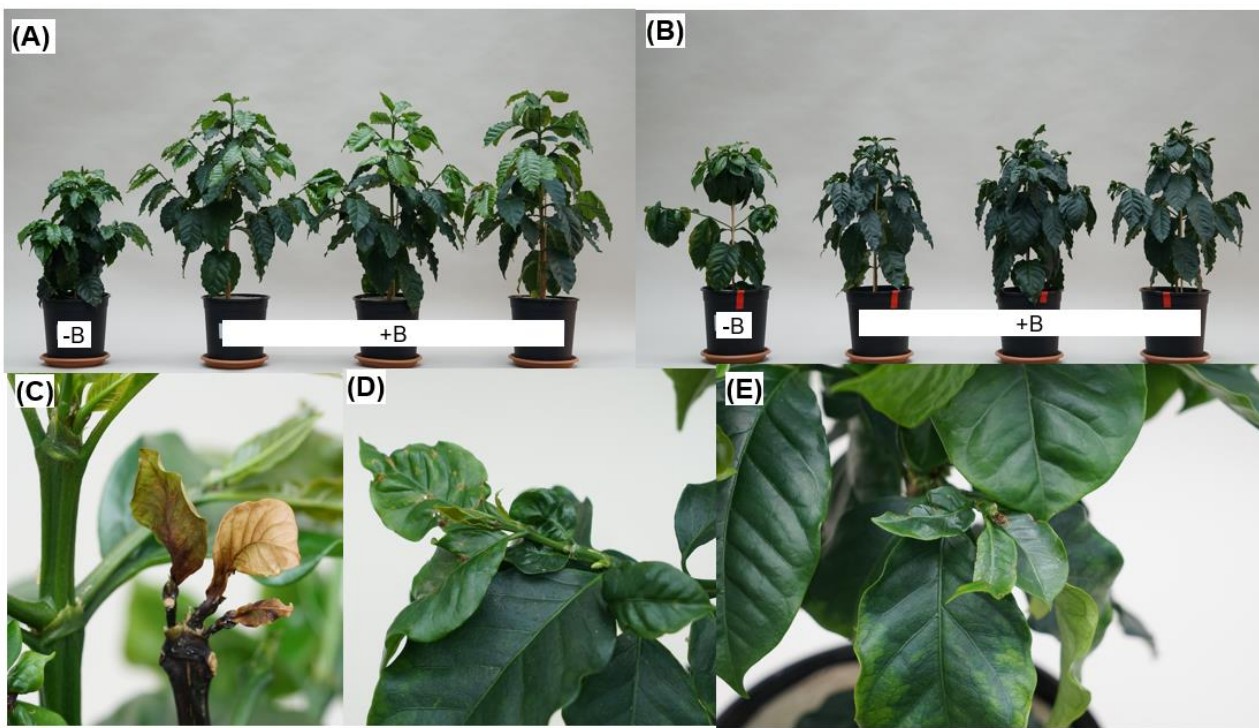

**Figure 2.** Treatments without water stress with and without boron (**A**). Treatments with water stress with and without Boron, (**B**). B deficiency symptoms (**C–E**).

**Table 4.** Influence of boron and water deficit on the nitrogen and boron contents in the leaves and uptake by the coffee plants nine months after transplanting under greenhouse conditions.

| Water Level (WL) | B Level (BL) | Nutrient Content | | | | Nutrient Uptake by Plants | |
|---|---|---|---|---|---|---|---|
| | | Young Leaves | | Mature Leaves | | N | B |
| | | N (%) | B (mg.kg$^{-1}$) | N (%) | B (mg.kg$^{-1}$) | mg. plant$^{-1}$ | |
| Without Water Stress | +B | 3.12 a | 63.72 b | 2.41 a | 265.07 c | 1873.4 b | 11.63 c |
| | −B | 3.70 b | 12.73 a | 3.72 b | 9.56 a | 1553.6 a | 6.32 b |
| With Water Stress | +B | 3.37 ab | 249.14 c | 3.86 b | 80.54 b | 1413.3 a | 0.38 a |
| | −B | 4.69 c | 11.44 a | 4.27 c | 11.93 a | 1402.8 a | 0.37 a |
| WL | | ** | ** | ** | ** | ** | ** |
| BL | | ** | ** | ** | ** | ** | ** |
| WL × BL | | * | ** | ** | ** | * | ** |

Different letters denote statistically significant differences according to Tukey's test, alpha = 0.05. * *p*-value < 0.005; ** *p*-value < 0.01; LSD for N in young leaves = 0.4458; for B in young leaves = 49.62577; for N in mature leaves = 0.23339; for B in mature leaves = 17.28739; and LSD for N uptake = 180.69550 and for B uptake = 0.99027.

### 3.2. B Influence on the Coffee Yield under Field Conditions

In the field trial, the water availability during the growing season from 2015 to 2019 was variable. In the season of 2015–2016, there was a low net rainfall and low ETc compared to the other years, indicating a longer drought stress; the period of 2016–2017 showed a higher net rainfall and ETc (Table 5). The net rainfall and ETc variation between production seasons explain the large and significant yield variation among the years (Figure 3).

During the first three years of harvest (from 2016 to 2018), no significant differences in the fresh coffee cherry yield among the treatments were observed. However, the treatments with the highest rate of B (1.1 kg B.ha$^{-1}$) produced more coffee compared to the treatment with 0.6 kg B.ha$^{-1}$ and the control without Ca$^{2+}$ and B. After three years of treatment, in 2019, significant differences were observed among the treatments. The treatment with 77 kg CaO.ha$^{-1}$ and 1.1 kg B.ha$^{-1}$ showed a coffee yield of 18,607 kg of cherries.ha$^{-1}$,

which was 32% higher than the treatment with 0.6 kg B.ha$^{-1}$ with 14,083 kg of cherries.ha$^{-1}$ and 93% higher than the treatment without CaO and B with 9615 kg of cherries.ha$^{-1}$ (Figure 3).

**Table 5.** Water balance parameter of the trial from pre-flowering to harvest each year (May to May).

| Period | Harvest Year | Net Rainfall | Crop Evapotranspiration (ETc) |
|---|---|---|---|
| | | | mm |
| 2015–2016 | 2016 | 529.0 | 565.0 |
| 2016–2017 | 2017 | 1096.0 | 980.0 |
| 2017–2018 | 2018 | 925.0 | 901.0 |
| 2018–2019 | 2019 | 837.0 | 833.0 |

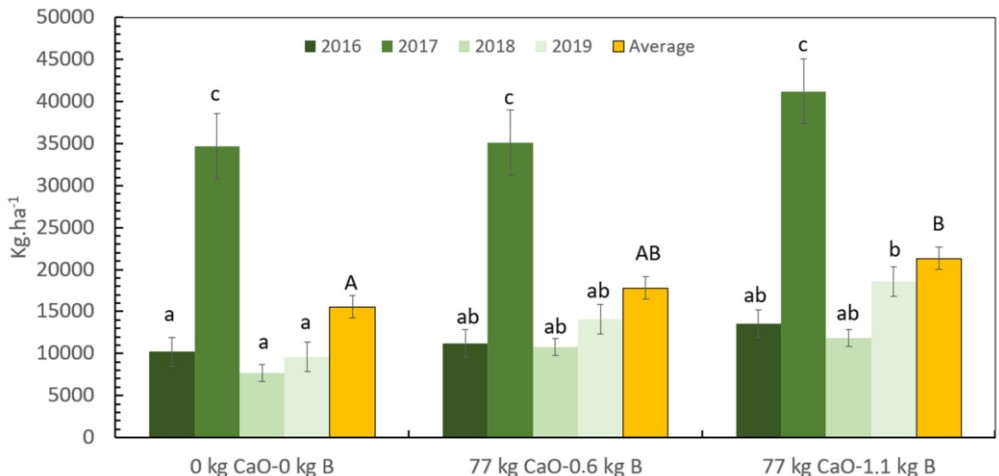

**Figure 3.** Influence of B rates in synergy with Ca on the long-term coffee yield. Vertical bars indicate the standard error and different letters have significant differences according to Tukey's test, alpha value of 0.05. The uppercase letters represent the differences in the mean four harvests' yield, and the lowercase letters represent the differences in the yield analyzed independently. LSD for 2016 = 5002.53, 2017 = 11,648.5, 2018 = 3124.2, 2019 = 4362.1, and for the average = 5689.1.

At the end of the productive cycle after 5 years of treatment application and 4 years of harvest, the treatment with the highest doses of B (1.1 kg.ha$^{-1}$) and Ca$^{2+}$ (77 kg.CaO.ha$^{-1}$) produced 37% more coffee cherries compared to the control without Ca$^{2+}$ and B over 5 years. The treatment with a lower B rate (0.6 kg.ha$^{-1}$) showed a 19% yield reduction compared to the treatment with 1.1 kg B.ha$^{-1}$, but a 14% higher yield than the control treatment without Ca$^{2+}$ and B (not significantly different).

*3.3. B in the Soil after Five Years of Application in the Field Trial*

After 5 years of B application, the B content in the soil at a 0 to 30 cm depth significantly increased compared to the control without B (Figure 4). The B content in the soil increased from 0.28 mg B.kg$^{-1}$ in the control to 0.705 and 1.175 mg B.kg$^{-1}$ in the treatments with mean B rates of 0.6 and 1.1 kg.ha$^{-1}$.year$^{-1}$, respectively.

*3.4. B and the Biochemical Composition of the Green Coffee Beans Produced under Field Conditions*

The biochemical composition of the green coffee beans, specifically the caffeine content and fatty acids (FAs), like myristic acid (C14:0), oleic acid (C18:1), and linolenic acid (C18:3), were significantly affected by the interaction between treatments and years (Table 6). The coffee samples from the 2016 harvest showed higher concentrations of these compounds compared to the samples from the harvest in 2019. The 2019 samples from the treatment with Ca$^{2+}$ and a B rate of 1.1 kg.ha$^{-1}$.year$^{-1}$ had significantly lower oleate acid (C18:1) and

myristic acid (C14:0) contents, and the treatment with a lower B rate of 0.6 kg.ha$^{-1}$.year$^{-1}$ showed a significantly lower linolenic acid (C18:3) content. Significant and inverse correlations were observed between the B content of the green coffee beans and caffeine, myristic acid (C14:0), oleic acid (C18:1), linoleic acid (C18:2), and linolenic acid (C18:3), with correlation coefficients of −0.5684, −0.6944, −0.6298, −0.6344, and −0.5923, respectively (Table 7).

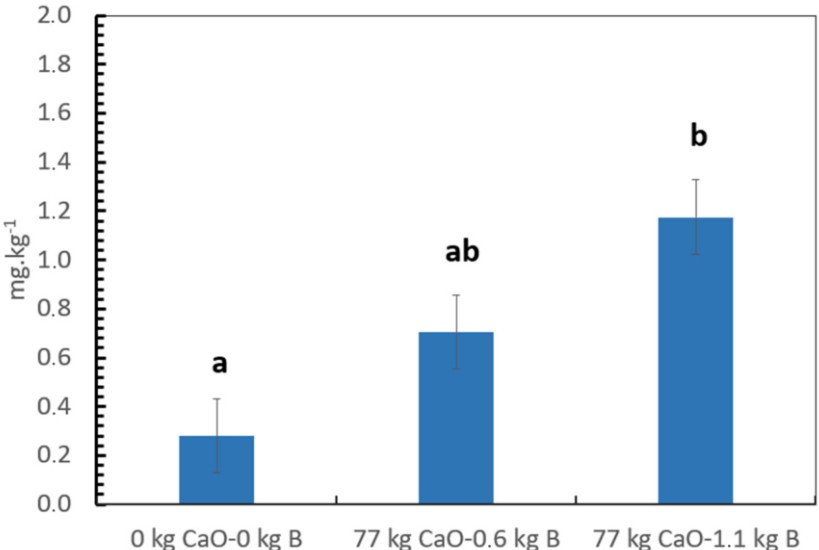

**Figure 4.** Changes in the B content in the soil after five years of treatment application. Vertical bars indicate the standard error and different letters show significant differences according to Tukey's test, alpha value of 0.05, and LSD = 0.50577.

**Table 6.** Influence of the B rates on the biochemical composition of the green coffee beans in the Caturra trial for two contrasting years.

| Treatment | Year | Caffeine | C14:0 Myristic | C16:0 Palmitic | C18:0 Stearic | C18:1 Oleate | C18:2 Linoleate | C18:3 Linolenate | C20:0 Arachidic | C22:0 Behenic |
|---|---|---|---|---|---|---|---|---|---|---|
| | | g.100g$^{-1}$ | | | | | | | | |
| 77 kg CaO-1.1 kg B | 2016 | 1.35 c | 0.06 c | 19.44 | 3.86 | 4.94 c | 44.56 c | 2.84 c | 1.27 | 0.35 |
| 77 kg CaO-0.6 kg B | | 1.36 c | 0.06 c | 20.47 | 3.98 | 5.09 c | 44.85 c | 2.88 c | 1.32 | 0.36 |
| 0 kg CaO-0 kg B | | 1.40 c | 0.05 c | 18.61 | 3.65 | 4.67 c | 41.32 bc | 2.66 c | 1.21 | 0.34 |
| Average | | 1.37 B | 0.06 B | 19.5 | 3.83 | 4.90 B | 43.58 B | 2.79 B | 1.27 | 0.35 |
| 77 kg CaO-1.1 kg B | 2019 | 1.11 b | 0.01 a | 16.00 | 3.15 | 0.47 a | 27.11 a | 2.11 b | 1.24 | 0.32 |
| 77 kg CaO-0.6 kg B | | 1.12 b | 0.03 b | 16.45 | 3.32 | 2.74 b | 29.12 a | 1.49 a | 1.29 | 0.34 |
| 0 kg CaO-0 kg B | | 1.02 a | 0.03 b | 19.78 | 3.89 | 2.19 b | 33.78 ab | 2.02 b | 1.53 | 0.41 |
| Average | | 1.09 A | 0.02 A | 17.4 | 3.45 | 1.80 A | 30.0 A | 1.88 A | 1.35 | 0.36 |
| Significance | | | | | | | | | | |
| Treatment | | NS | NS | NS | NS | * | NS | NS | NS | NS |
| Year | | ** | ** | NS | NS | ** | ** | ** | NS | NS |
| Treatment × Year | | * | * | NS | NS | * | NS | * | NS | NS |

Different letters denote statistically significant differences according to Tukey's test, alpha = 0.05. * *p*-value < 0.005; ** *p*-value < 0.01. LSD for caffeine = 0.09101, myristic = 0.01826, oleate = 12.01564, and linolenate = 0.76024. NS: not significantly different.

**Table 7.** Pearson correlation coefficients for the B content of the green coffee beans and the biochemical composition.

| Pearson Correlation | Caffeine | C14:0 Myristic | C16:0 Palmitic | C18:0 Stearic | C18:1 Oleate | C18:2 Linoleate | C18:3 Linolenate | C20:0 Arachidic | C22:0 Behenic |
|---|---|---|---|---|---|---|---|---|---|
| | | | | | g.100g$^{-1}$ | | | | |
| Correlation coefficient | −0.5684 | −0.6944 | −0.3401 | −0.3362 | −0.6298 | −0.6344 | −0.5923 | −0.0197 | −0.148 |
| *p*-value | 0.0047 ** | 0.0002 *** | 0.1123 ns | 0.1167 ns | 0.0013 ** | 0.0012 ** | 0.0029 ** | 0.9288 ns | 0.5004 ns |

*p*-value < 0.05 **; *p*-value < 0.01 ***; ns: not significantly different.

## 4. Discussion

*4.1. Influence of the Iteraction of Water and B Deficits on the Biomass Accumulation, B and N Uptake under Controlled Conditions*

In the treatments with adequate nutrition with B and without water stress, a decreasing acropetal gradient was observed in the B content between the mature and youngest leaves (Table 4). The decreasing acropetal gradient in the treatments without water stress and without B and with water stress and without B were reversed, changing to an increasing acropetal gradient, from 9.56 to 12.72 mg.kg$^{-1}$ in the treatment without water stress and without B and from 80.54 to 249.14 mg.kg$^{-1}$ in the treatment with water stress and without B. These results suggest that B translocation from the mature to young leaves can occur when there is a low B supply in plants grown with and without water deficit. This process was well described by Brown and Shelp [15] for several crops, like broccoli, soybean, cotton, grapes, and peanuts. In these crops, a decreasing acropetal gradient of the B concentration was found when grown with an excess or adequate B supply. However, the authors reported that, when these crops were grown under a B deficit, the gradient from old to young tissues disappeared or was reversed.

The changes in the B concentration between the mature and young leaves under water stress conditions suggest a remobilization of B under water stress conditions, which is not observed in the well-watered treatments (Table 4). Considerable evidence suggests that B can be transported to the sink tissues in the phloem of many species [15,45–47]. This B mobility in the phloem can occur in any species in which sorbitol is a primary photosynthesis product. Polyols, like sorbitol, mannitol, and dulcitol, can effectively complex B [48,49]. Mannitol is one of the most widely distributed polyols and is present in over 100 higher plants, including Rubiaceae, to which coffee belongs [49].

Leite et al. [46], utilizing isotopic B labeling, provided evidence that B remobilization occurred in coffee trees, with some translocation in well-nourished plants (without B deficit), but significant phloem translocation or remobilization at a deficient supply of B, as what was reported in this research for the treatments without drought stress and without B. Olivera-Silva et al. [50] demonstrated B remobilization in Cowpea (*Vigna unguiculata* (L.) Walp), which in turn resulted in a high B accumulation and a significant increase in the dry biomass accumulation. Bellato et al. [51] reported that more than 50% of the B contained in coffee leaves was found in the cell wall fraction, with the remaining B being soluble. This implies that the B fraction in coffee cells that is not complexed and is relatively high and hence may be available for remobilization. In citrus plants, Boaretto et al. [52,53] reported B remobilization from older tissues, estimating that 30–35% of the total boron in the leaves of a new flush were remobilized from plant reserves.

At both water levels (with and without drought stress), the N content in young and mature leaves was significantly higher in the treatments without B (Table 4). This could be explained by the influence of B on the N metabolism. The influence of B on N assimilation could be observed in the coffee plants that were grown with water deficit: The treatments with water stress and without B supply did not show significant differences in N uptake, but the N content in the young and mature leaves in the treatments with B supply was significantly lower (Table 4) and the biomass accumulation was significantly higher

(Figure 1). Wang et al. [54] reported a significant improvement in N uptake, NUE, and yield in rapeseed (*Brassica napus*) with the application of a B fertilizer, with a higher benefit of the B application on higher N rates (180 kg N.ha$^{-1}$) than in the treatments without and/or lower N rate.

The way in which these micronutrients affect drought sensitivity in plants can be explained in two ways. Firstly, B together with Zn and Mn are involved in the detoxification of ROS, playing a protective role in preventing the photo-oxidative damage catalyzed by ROS in chloroplasts. Secondly, these micronutrients may greatly contribute to drought stress tolerance by protecting membranes against oxidative damage [55,56]. The levels of ascorbic acid, non-protein SH compounds (mainly glutathione), and glutathione reductase, the major defense systems of cells against toxic $O_2$ species, are reduced in response to B deficiency [56]. Yan et al. [57], working with citrus, grown with and without acid stress of "excessive $H^+$ protons at pH of 4.0", demonstrated that the application of B protected plant roots from $H^+$-toxicity by inhibiting the outbreak of ROS in the roots and regulating the protective mechanism of antioxidant enzymes.

Boron is essential for organisms with carbohydrate-rich cell walls, and symptoms of B deficiency include a cessation in the growth of apical meristems (both shoots and roots) and the development of the brittleness of leaves, which has been ascribed to an inhibition of cell wall synthesis or structural integrity. Boron deficiency also results in the formation of abnormally thick and structurally deformed cell walls [24]. The most frequent B deficit symptoms observed in this research were: the deformation and the presence of brown spots on the youngest leaves. When the leaves reached a month of development, discoloration appeared on the outer edge of the leaves. The most severe symptom of the B deficit was a descending branch death or the cessation in the growth of the apical shoot meristems and the necrosis of the tissues (Figure 2B). This descending branch death has been associated with environmental stresses, such as soil and atmospheric water deficits, high temperature, high insolation, N deficits, or a combination of all [58]. In the juvenile phase of coffee growth, the death of the branches does not represent a problem; it is, however, frequently seen in productive plants [58]. We observed this physiological disorder also in juvenile plants growing without B. Branch death was also often observed in the field trial after 3 years of the treatment without $Ca^{2+}$ and B, which also explained the significant differences in productivity at the end of the productive cycle (Figure 1).

The optimum foliar B concentration for coffee in the third and fourth pair of leaves is considered normal between 60 and 80 mg.kg$^{-1}$ [21,59]. In this research, in well-watered conditions, the highest biomass accumulation was achieved with a B content in the young leaves of 63.72 mg.kg$^{-1}$ and in mature leaves of 265.07 mg.kg$^{-1}$. However, under water stress conditions, due to the remobilization of B, the highest biomass accumulation was achieved with a B content in young leaves of 249.14 mg.kg$^{-1}$ (without toxicity symptoms) and 80.54 mg.kg$^{-1}$ in mature leaves (Table 7).

### 4.2. Influence of B on Coffee Productivity

The yield in 2017 was high in all treatments (Figure 3), because the coffee plantation was young (3 years after pruning) and because of the high net precipitation and ETc during the period of 2016–2017. The net rainfall in 2016–2017 was almost two times higher, and the ETc was 73% higher compared to the previous period of 2015–2016 (Table 5). The period from 2015 to 2016 was influenced by the Pacific South Oscillation ENSO (El Niño-La Niña) conditions, which were in a positive phase known as El Niño [60,61]. El Niño for the study area resulted in an increase in the mean air temperature, an increase in solar radiation (Table 1), and a reduction in the rainfall. For the coffee plantation, this reduction in rainfall during tree growth and coffee cherry development results in a decline in the growth, yield, and quality of the coffee [5,8]. A stress period after harvest and during pre-flowering, however, has a positive influence on productivity through a better flower induction [62,63].

The application of an average rate of 77 kg CaO and 1.1 kg.B.ha$^{-1}$.year$^{-1}$ improves the coffee yield in the 4th year of application (Figure 3) and the B content in the soil (Figure 4),

compared to the treatment with the same $Ca^{2+}$ rate and with less B (0.6 kg.B.ha$^{-1}$.year$^{-1}$) and the control without both nutrients, indicating the importance of soluble $Ca^{2+}$ and B in coffee in the long term. Boron nutrition in crops is a challenging practice, due to the high mobility of B in many soils, its high adsorption capacity, and the narrow window between deficiency and toxicity [64–66]. In Arabica coffee in Brazil, Santinato et al. [59] found that the application of B at a rate higher than 2.0 kg.B.ha$^{-1}$ linearly reduced the yield by 0.33 coffee bags.ha$^{-1}$ (0.02 t of green coffee beans.ha$^{-1}$) per kg of B applied in excess of 2.0 kg.B.ha$^{-1}$. In contrast Cong [67], working on a 10-year-old Robusta coffee plantation during one season in a highly acidic soil in Vietnam (basaltic soil of the Central Highlands), reported an average yield increase of 10.2% with the application of 3.0 kg B.ha$^{-1}$. These results indicate that the response to B nutrition is variable according to the coffee species and the environment.

The additive effect between B and $Ca^{2+}$ after long-term application observed in the field trial can also be explained by the influence of both nutrients on the cell structure of coffee plants. Both nutrients have been reported as key nutrients in coffee plants, influencing metabolic structural and morphological processes [32,35]. B acts in the biosynthesis of cell walls, assisting $Ca^{2+}$ in the deposition of pectates in cell walls. Boron also forms cis-diol-borate complexes, which are constitutional elements of the plasmalemma [50].

The interaction between B and $Ca^{2+}$ was described by Brown et al. [24], in which $Ca^{2+}$ and B play a cooperative role in the stabilization of the membrane by the formation of a mixed complex in which $Ca^{2+}$ binds to polyhydroxyl borate esters or by direct association with different compounds of the membrane. In *Vicia faba* roots, the membrane-bound $Ca^{2+}$ decreased within several hours of B-deficient conditions due to the reduction in specific $Ca^{2+}$-binding sites (borate esters with vic-diols or polyhydroxil-carboxylates) before the plasma membrane integrity deteriorates [68]. Synergies between $Ca^{2+}$ and B and their influences on crop productivity also have been reported in a 3-year trial in cranberries by DeMoranville and Deubert [69]. The authors found a yield increase from 24 to 31% with foliar $Ca^{2+}$ and B applications during the period of strong vegetative growth and floral development, and recently, Galeriani et al. [70] reported a significant improvement in gas exchange parameters, water use efficiency, and yield in soybean.

Malavolta et al. [33] proposed, for B in soils, a low level below 0.4 mg.kg$^{-1}$, a medium level between 0.4 and 0.8, and higher levels when B is higher than 0.8 mg.kg$^{-1}$. In our trial, higher yields were achieved with a B level in the soil of 1.17 mg.kg$^{-1}$ in the soil using the CAT extraction methods (with a confidence interval from 0.86 to 1.49 mg.kg$^{-1}$). The treatment without B and $Ca^{2+}$ showed a significantly lower coffee yield after four years of harvest and a low B content in the soil of 0.28 mg.kg$^{-1}$, much lower than the B content at the beginning of the experiment (0.38 mg.kg$^{-1}$).

### 4.3. B and the Biochemical Composition of the Green Coffee Beans under Field Conditions

The chemical characteristics of green coffee beans are determined by the combination of three main factors: environmental x genetic x agricultural practices [71]. The chemical components of roasted coffee can be grouped into volatile and non-volatile substances. Non-volatile compounds, such as caffeine and lipids, together with sugars, trigonelline, and chlorogenic acids, influence the beverage quality [72,73]. Caffeine has negative and significant correlations with all cup quality attributes of coffee, such as acidity, body, flavors, and overall standard of the liquor [74], and is mostly associated with the bitterness in the coffee cup [72,75]. Fatty acids, as constituents of the lipid fraction, contribute to the aroma, flavor, and mouth feel perceived, which are usually influenced by the type and concentration of lipids [73]. Saturated fatty acids, including arachidic (C20:0), stearic (C18:0), and palmitic acid (C16:0), have been described as potential discriminators of specialty coffees, indicating a better sensory quality. In contrast, unsaturated fatty acids (UFAs), such as oleic (C18:1), linoleic (C18:2), and linolenic (C18:3) acids, have been related to a coffee beverage with less intense acidity, fragrance, and flavor [76] and have also been linked to a loss of sensory quality during storage [77].

The significant differences in caffeine, and some fatty acids, specially C18:1; C18:2, and C18:3, between 2016 and 2019, is explicable by the changes in rainfall and ETc generated by the ENSO conditions. During the period of 2015–2016, corresponding to the harvest of 2016, the climate conditions were warmer and drier compared to the period of 2018–2019, with 58% less net rainfall and a 47% lower ETc (Table 5). The water stress and foliar B application can alter the seed composition in soybeans [78], with an increase in the oleic acid concentration under water stress compared to non-stress conditions, without changes in the linolenic acid concentration. In both conditions, with and without stress, the foliar B application resulted in an increase in oleic acid and in a decrease in the linolenic content.

Fatty acids and mainly UFAs are recognized as important in the general defense system against various biotic and abiotic stresses [79]. In plants, fatty acids (FAs) are crucial components of cellular membranes and suberin and cutin waxes that provide structural barriers to the environment and contribute to inducible stress resistance through the remodeling of membrane fluidity and as modulators of plant defense gene expression [80]. DaMatta and Ramalho [4] associated the better acclimation of some coffee genotypes to abiotic stress conditions (cold stress) with the increase in the saturation of FAs in the membrane that reinforced the antioxidative system. Wu et al. [81] demonstrated that B deficiency causes structural and morphological changes in trifoliate oranges roots. B deprivation-induced ROS accumulation accelerated the membrane peroxidation, resulting in a weakened cell vitality and cell rupture in roots. Chen et al. [82] found that B deficiency led to an excessive accumulation of phenolic compounds in alfalfa seeds. In this paper, we reported a negative correlation between caffeine and B content in green coffee beans (Table 7)

Boron deficiency caused an increase in the cytosolic calcium concentration ($Ca^{+2}_{cyt}$) in *Arabidopsis thaliana* roots after 6 and 24 h of this nutrient deficiency. The ($Ca^{+2}_{cyt}$) was gradually restored with B application or when B sufficiency conditions were established [83].

## 5. Conclusions

After a seven-month trial under controlled conditions and a five-year trial under field conditions, it is possible to conclude that:

— Boron deficits significantly reduce the growth of the young coffee plants, influencing mainly the shoot biomass accumulation and clearly indicating the strong susceptibility of the coffee plants to B deficit.
— Coffee plants are sensitive to long water stress periods, and B application significantly reduces the negative effect of this abiotic stress.
— Under water stress conditions, it was possible to observe B remobilization in the phloem from mature to young leaves.
— During a whole production cycle of five years, the mean application of 77 kg CaO and 0.6 kg B.ha$^{-1}$.year$^{-1}$ yielded 14% more than the control without B and $Ca^{2+}$, while the increase in the mean B rate to 1.1 kg.ha$^{-1}$.year$^{-1}$ at the same $Ca^{2+}$ rate increased the productivity by almost 37% compared to the control, indicating a long-term additive effect between B and $Ca^{2+}$.
— Unsaturated fatty acid and caffeine contents in the green coffee beans were good indicators of the stress conditions (water stress) under field conditions. These compounds were negatively correlated with the B content, indicating that B could reduce the formation of these compounds, which are considered as negative descriptors of coffee cup quality.

**Author Contributions:** Conceptualization, data analysis, and paper preparation, V.H.R.-B. and J.K.; greenhouse trial implementation and data acquisition, E.T.; field trial implementation and data acquisition, V.H.R.-B. and L.A.L.-V. All authors have read and agreed to the published version of the manuscript.

**Funding:** This research was supported by Yara International.

**Data Availability Statement:** Data are contained within the article.

**Acknowledgments:** The authors acknowledge the assistance with the biochemical analysis from the National University of Colombia, Medellin Campus, Food Science Department; Elias Roa and Tim Wendelboe, coffee producers, who supported the field trial for five years; and the external reviewers, for their contributions and suggestions, which helped to improve the manuscript.

**Conflicts of Interest:** Author Luis Alfredo Leal-Varon was employed by the company Yara Colombia. The remaining authors declare that the research was conducted in the absence of any commercial or financial relationships that could be construed as a potential conflict of interest.

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
