# Peer review of "Boron Nutrition in Coffee Improves Drought Stress Resistance and, Together with Calcium, Improves Long-Term Productivity and Seed Composition"

_agronomy, doi:10.3390/agronomy14030474_

Round 1

Reviewer 1 Report

Comments and Suggestions for Authors

Review of agronomy-2889998 “Boro Nutrition in Coffee Improves Drought Stress Resistance and in Interaction with Calcium Improves Long-Term Productivity”

This study addresses boron nutrition and its interaction with calcium in coffee, based on a combination of a short-term greenhouse experiment and 5-year field experiment. The study shows strong effects of boron deficiency, including yield increases in the field in Colombia. The paper also presents data on treatment effects on unsaturated fatty acids that are important indicators of coffee bean quality, with the general result being improved quality in response to B and Ca additions.

In general, this is a solid study and contributes new information with good potential for application on the important but often neglected nutrient boron in coffee cultivation. The combination of greenhouse and long-term field makes for a particularly compelling results, and the data on UFAs are also important.

There is strong support for the main results, but minor improvements in statistical analysis are needed. The discussion also addresses possible “synergy” in B and Ca effects, but there are no formal analyses of statistical interactions of B and Ca. The post-hoc tests used are also problematic (Fisher’s LSD test is not appropriate for more than 3 groups). I don’t any main conclusions (other than statements on “synergy”) need to be substantively changed. Thus, although there are many small corrections, I consider the necessary revisions “minor”.

Main criticisms:

1.     “SYNERGY” BETWEEN B AND CA? The heading of section 4.2 reads “Sinergy Ca-B on coffee productivity” and implies that an “interaction” between B and Ca was found in the experiments. Confusingly, this section also states that the effects of B and Ca are “additive”.

This section and other statements on interactive effects of B and Ca on coffee performance needs to be substantially revised. Neither experiment involved a factorial combination of B and Ca treatments, so the interactions between these cannot be assessed. “Synergy” is used loosely and not defined – but should strictly refer to positive deviations from additivity in the context of a factorial experiment.

2.     POST-HOC TESTS. The Fisher LSD procedure only preserves the overall type I error rate where there are three groups; in general, the Tukey HSD procedure or similar should be used. This needs to be corrected.

3.     CAT SOLUTION EXTRACTION OF B. Line 141. Description of CAT solution is not clear (1:10 ratio?). Also, a more complete description of the extraction procedure is needed.

4.     SHOULD SHOW PHOTOS OF MAIN SYMPTOMS OF B DEFICIENCY DESCRIBED. Lines 376-388 describe gross morphology symptoms of B deficiency based on the experiments. These are very useful as a reference to growers. Photos should be included to document these generally and for applied use.

5.     CRITICAL B LEVELS. Section 4.3 in general needs revision; it addresses critical thresholds for B deficiency, but only states the B levels in soil that resulted in increased yields based on experiments. The data available potentially would allow for estimation of critical levels if individual performance can be matched to soil levels. The final statements on recommended soil levels of B are also not supported without such an analysis. In addition, critical thresholds for nutrients are most often assessed using foliar levels. It is surprising that this is not mentioned at all here.

6.     NEED FOR REVISION FOR CORRECT TERMINOLOGY USE, ETC. The paper requires extensive corrections in general – even the first word of the title is incorrect. Manyt corrections are noted on the marked ms, but a a few specific recurring points:

a.     The correct specification for ion charges should be used (e.g., Ca^2+ not Ca^+2)

b.     English-language decimal points should be used (e.g., 0.6 not 0,6).

c.     Line 176. What are “effective plants”?

d.     Line 371. “monachism” incorrectly used.

Comments on the Quality of English Language

As noted in review and marked file, minor corrections are needed.

Author Response

Reviewer 1:

Review of agronomy-2889998 “Boro Nutrition in Coffee Improves Drought Stress Resistance and in Interaction with Calcium Improves Long-Term Productivity”

This study addresses boron nutrition and its interaction with calcium in coffee, based on a combination of a short-term greenhouse experiment and 5-year field experiment. The study shows strong effects of boron deficiency, including yield increases in the field in Colombia. The paper also presents data on treatment effects on unsaturated fatty acids that are important indicators of coffee bean quality, with the general result being improved quality in response to B and Ca additions.

In general, this is a solid study and contributes new information with good potential for application on the important but often neglected nutrient boron in coffee cultivation. The combination of greenhouse and long-term field makes for a particularly compelling results, and the data on UFAs are also important.

There is strong support for the main results, but minor improvements in statistical analysis are needed. The discussion also addresses possible “synergy” in B and Ca effects, but there are no formal analyses of statistical interactions of B and Ca. The post-hoc tests used are also problematic (Fisher’s LSD test is not appropriate for more than 3 groups). I don’t any main conclusions (other than statements on “synergy”) need to be substantively changed. Thus, although there are many small corrections, I consider the necessary revisions “minor”.

 Answer: All the post-hoc test was changed for Tukey test. Our experimental design in the field trial does not allow us to have an interaction between B and Ca, we just have one treatments with low B and other with high B at the same rate of CaO and one control without Ca and B. The intention of the trial at the moment of the implementation was only compared the effect of the B reduction (aprox 50%) and compare with a control without both. In the discussion, the synergy or interaction is changed by the additive concept more than Synergy.  

Main criticisms:

  1. “SYNERGY” BETWEEN B AND CA? The heading of section 4.2 reads “Sinergy Ca-B on coffee productivity” and implies that an “interaction” between B and Ca was found in the experiments. Confusingly, this section also states that the effects of B and Ca are “additive”.

This section and other statements on interactive effects of B and Ca on coffee performance needs to be substantially revised. Neither experiment involved a factorial combination of B and Ca treatments, so the interactions between these cannot be assessed. “Synergy” is used loosely and not defined – but should strictly refer to positive deviations from additivity in the context of a factorial experiment.

  Answer: Our experimental design in the field trial does not allow us to have an interaction between B and Ca, we just have one treatment with low B and other with high B at the same rate of CaO and one control without Ca and B.

The main objective of the trial at the moment of the implementation was only compared the effect of the B reduction (aprox 50%) and compare with a control without Ca and B. In the discussion, the synergy was corrected and changed. 

  1. POST-HOC TESTS. The Fisher LSD procedure only preserves the overall type I error rate where there are three groups; in general, the Tukey HSD procedure or similar should be used. This needs to be corrected.

Answer: All the post-hoc test was changed for Tukey test.

  1. CAT SOLUTION EXTRACTION OF B. Line 141. Description of CAT solution is not clear (1:10 ratio?). Also, a more complete description of the extraction procedure is needed.

Answer: We include a reference that explain in detail the CAT method: Baumgarten, A.; Dachler, M. The assessment of nutrient availability in soils using the CaCl2/DTPA (CAT) extraction method. Acta Hortic. 2000,511,35-42.doi: 10.17660/ActaHortic.200.511.3 

  1. SHOULD SHOW PHOTOS OF MAIN SYMPTOMS OF B DEFICIENCY DESCRIBED. Lines 376-388 describe gross morphology symptoms of B deficiency based on the experiments. These are very useful as a reference to growers. Photos should be included to document these generally and for applied use.

 Answer: Tree new photos include into the figure 2.(Fig, 2C,D,E).

  1. CRITICAL B LEVELS. Section 4.3 in general needs revision; it addresses critical thresholds for B deficiency, but only states the B levels in soil that resulted in increased yields based on experiments. The data available potentially would allow for estimation of critical levels if individual performance can be matched to soil levels. The final statements on recommended soil levels of B are also not supported without such an analysis. In addition, critical thresholds for nutrients are most often assessed using foliar levels. It is surprising that this is not mentioned at all here.

Answer: The section 4.3 was deleted due to lack of data to estimate a critical threshold for B in this trial. Regarding with the foliar, we do not sampling consistently years by year, so no enough data to include in the results.

  1. NEED FOR REVISION FOR CORRECT TERMINOLOGY USE, ETC. The paper requires extensive corrections in general – even the first word of the title is incorrect. Manyt corrections are noted on the marked ms, but a a few specific recurring points:

  1. The correct specification for ion charges should be used (e.g., Ca^2+ not Ca^+2) Answer: Done

  1. English-language decimal points should be used (e.g., 0.6 not 0,6). Answer: Done
  2. Line 176. What are “effective plants”? Answer: each experimental unit has a plot with  28 plants, but only 10 were selected for register the harvest and take the beans and soil samples, the other 18 were the borders.

  1. Line 371. “monachism” incorrectly used. Answer: Done

Reviewer 2 Report

Comments and Suggestions for Authors

Dear editor and authors, the manuscript is good and deserves to be published. However, I suggest some adjustments:

All text

- Check the spelling, grammar and punctuation of the text, there are several small mistakes.

- Lack of standardization between the use of acronyms or the entire text example (N, nitrogen).

Abstract

It's ok, and it contains the essence of the study!

Introduction

- Provide correct access to information for (Yara International, Megalab data base – line 53), as this information only appears in an unavailable database, and may be speculative.

- Start sentences with the full text and not acronyms when applicable, example line 55, line 60.

- The introduction lacks a paragraph showing how boron can mitigate water stress, focus on work. There is several studies with other crops that already attest to this effect, highlighting the metabolic process involved.

- Authors must change the way they write the last paragraph of the introduction, making the hypothesis and aims of the study clear.

Material and Methods

- In lines 102 and 121, cite the sources, i.e. the list of reagents used, as well as their degree of purity.

Results

- In Figure 1 and table 4, the authors have an experimental scheme as a 2 x 2 factorial (application of B x stress), because the authors did not make the comparison in this scheme, and simply ignored a possible interaction effect between the factors studied ? The most appropriate approach would be this, because it would in fact be clearer to answer one of the study's hypotheses.

- In figure 2, why are there three plants to represent the same treatment? – I suggest redoing the figure, placing one representative plant per treatment, and placing the 4 plants (4 treatments) in the same figure, it will be clearer for the readers to see the differences.

- Table 5 is not a result, it is part of the material and methods, it is easy to verify because there is no statistical test in the comparison presented. Therefore, table 5 and the information in its text should be moved to the material and methods section.

- In Figure 3 and table 6, how do the authors compare treatments and years together? Because this is not part of an experimental design, this is statistically incorrect, I suggest redoing the comparison of treatments each year, and at the end making a figure with the average comparison. The comparison of years as is done would require another approach, perhaps the adoption of a comparison based on a split-plot scheme over time.

- In table 7, the captions used must be better described

Discussion

- I suggest making only two discussion topics, one focused on each experiment, it makes sense!

- The discussion can be more concise by avoiding repeating the results in study values, as they are already present in the results section.

Conclusions

- The second paragraph needs to be rewritten, it looks like a summary of the results section.

- After inserting the hypothesis and aims of the work more clearly, the conclusion must be modified to meet the aims of the study.

Author Response

Reviewer 2:

All text

- Check the spelling, grammar and punctuation of the text, there are several small mistakes. Answer: Done

- Lack of standardization between the use of acronyms or the entire text example (N, nitrogen). Answer: Done

Abstract

It's ok, and it contains the essence of the study!

Introduction

- Provide correct access to information for (Yara International, Megalab data base – line 53), as this information only appears in an unavailable database, and may be speculative.

Answer: That data base is internal for our company, we are not able to share.

- Start sentences with the full text and not acronyms when applicable, example line 55, line 60.

Answer: Done

- The introduction lacks a paragraph showing how boron can mitigate water stress, focus on work. There is several studies with other crops that already attest to this effect, highlighting the metabolic process involved.

Answer: New references were included from the line 71 to 81

- Authors must change the way they write the last paragraph of the introduction, making the hypothesis and aims of the study clear.

Answer: Done

 Material and Methods

- In lines 102 and 121, cite the sources, i.e. the list of reagents used, as well as their degree of purity.

 Answer: The nutrient solution used in the trial, is the results of our experience on greenhouse crop growth, in the specific case, several internal trials were conducted to develop the optimum nutrient rates for coffee, that is way we do not include any reference as a Hougland solution etc. For the nutrient solution we use pure nutrients with a purity between 95 to 99%.

Results

- In Figure 1 and table 4, the authors have an experimental scheme as a 2 x 2 factorial (application of B x stress), because the authors did not make the comparison in this scheme, and simply ignored a possible interaction effect between the factors studied ? The most appropriate approach would be this, because it would in fact be clearer to answer one of the study's hypotheses.

 Answer: The trial has a 2 x 2 factorial design and the analysis include the interaction between both factors Water Level (WL) and Boro Level (BL) WLxBL

- In figure 2, why are there three plants to represent the same treatment? – I suggest redoing the figure, placing one representative plant per treatment, and placing the 4 plants (4 treatments) in the same figure, it will be clearer for the readers to see the differences.

Answer: Unfortunately, the photos were made in this way, no photos with single plants.  

- Table 5 is not a result, it is part of the material and methods, it is easy to verify because there is no statistical test in the comparison presented. Therefore, table 5 and the information in its text should be moved to the material and methods section.

Answer: I prefer let as a part of the results, because I’m using as a key argument to explain the differences on productivity and coffee seed composition. Normally in climate data we do not have replications mainly because that rainfall is measure with only one rain gauge on the study area.  

- In Figure 3 and table 6, how do the authors compare treatments and years together? Because this is not part of an experimental design, this is statistically incorrect, I suggest redoing the comparison of treatments each year, and at the end making a figure with the average comparison. The comparison of years as is done would require another approach, perhaps the adoption of a comparison based on a split-plot scheme over time.

Answer: Done

- In table 7, the captions used must be better described

 Answer: Done

Discussion

- I suggest making only two discussion topics, one focused on each experiment, it makes sense!

Answer: The discussion was simplified to 3 main topics:

4.1. Interaction water and B deficits on biomass accumulation, B and N uptake under controlled conditions.

4.2. Influence of the B on coffee productivity.

4.3. B and the biochemical composition of the green coffee beans in field conditions.

- The discussion can be more concise by avoiding repeating the results in study values, as they are already present in the results section.

 Answer: Done

Conclusions

- The second paragraph needs to be rewritten, it looks like a summary of the results section.

- After inserting the hypothesis and aims of the work more clearly, the conclusion must be modified to meet the aims of the study.

 Answer: Done